# Temporomandibular Joint and Cervical Spine Mobility Assessment in the Prevention of Temporomandibular Disorders in Children with Osteogenesis Imperfecta: A Pilot Study

**DOI:** 10.3390/ijerph18031076

**Published:** 2021-01-26

**Authors:** Kulesa-Mrowiecka Małgorzata, Pihut Małgorzata, Słojewska Kinga, Sułko Jerzy

**Affiliations:** 1Department of Physiotherapy, Faculty of Health Science, Institute of Physiotherapy, Medical College, Jagiellonian University, 12 Michalowskiego Str., 31-143 Krakow, Poland; kinga.slojewska@uj.edu.pl; 2Prosthodontic Department, Faculty of Medicine, Institute of Dentistry, Medical College, Jagiellonian University, 4 Montelupich Str., 31-155 Krakow, Poland; malgorzata.pihut@uj.edu.pl; 3Faculty of Medicine, Orthopedic Clinic of the University Children’s Hospital, Medical College, Jagiellonian University, 265 Wielicka Str., 30-663 Krakow, Poland; jerzysulko@hotmail.com

**Keywords:** temporomandibular disorders, osteogenesis imperfecta, cervical spine, orofacial pain, differential diagnosis

## Abstract

Osteogenesis imperfecta is a heterogeneous group of hereditary disorders of connective tissue diseases characterized by increased bone fragility, low growth, sometimes accompanied by abnormalities within the dentine, blue sclera, and partial or total hearing impairment. The changes may affect all joints, including the cervical spine and temporomandibular joints in the future, resulting in pain. The aim of the present study was to assess whether there is a relationship between the active range of motion of the cervical spine and the mobility of temporomandibular joints due to differential diagnosis in children with osteogenesis imperfecta, and to present a prevention algorithm for temporomandibular disorders. The study involved a group of 34 children with osteogenesis imperfecta, and the control group included 23 children (age 9.1 ± 3.8 years). Data were collected through an interview based on the author’s questionnaire, and the physical examination consisted in measuring the mobility of the cervical spine using an inclinometer (Cervical Range of Motion Instrument), the Helkimo scale, and linear measurements. In children with congenital bone fragility, there were acoustic symptoms from the temporomandibular joints more often than in healthy children. A correlation was found between the mobility of the cervical spine and temporomandibular joints in the study group. In the case of detecting irregularities in the temporomandibular joints, children were ordered to perform jaw-tongue coordination exercises.

## 1. Introduction

Osteogenesis imperfecta (OI) is a group of genetic disorders of connective tissue [1], characterized by increased bone friability, low growth, and joint laxity, which can be accompanied by teeth and occlusion irregularities, blue ocular sclera, and hearing deficiencies or loss. The incidence rate of OI is 1 per 10,000–20,000 births [2] and does not depend on race or geographic location [3]. The most common cause of the disease is the mutation in the genes encoding alpha1 and alpha2 chains of type I collagen (COL1A1 and COL1A2), which results in quantitative or qualitative disorders in the collagen structure, which is the main structural protein of bones and dentine [4].

Inheritance of OI is typically autosomal dominant (90%), and the mutation can be inherited from one of the parents or it may appear in a child “de novo”. Symptoms of the disease can manifest in all tissues rich in type I collagen (i.e., bones, cords, joints, skin, sclera, inner ear) [1,4,5].

Osteogenesis imperfecta can be also caused by modifications in the CRTAP, LEPRE1, and PPIB genes, which are responsible for posttranslational modification of precursors of collagen and other factors participating in bone formation [4]. In this case, the disease is inherited autosomal recessive (10%). As a result of OI, the balance between the action of osteoblasts and osteoclasts is disrupted and leads to increased resorption of bones by osteoclasts [6]. Fractures heal easily, yet in about 60% of individuals they lead to bone deformities, which impede or entirely disable independent walking [4,7].

Extra-skeletal symptoms include blue sclera (in about 80% of individuals), caused by thinning and greater lucency, susceptibility to hemorrhages resulting from vascular friability, cranial deformities, and hearing disturbances or loss [4]. Furthermore, 50% of individuals with OI have improper dentine formation, known as dentinogenesis imperfecta. Despite the correct density and thickness of enamel, it is rapidly chipped, and the dentine is abraded [8,9]. Another common symptom are malocclusions, resulting from the inadequate head proportion to the rest of the body [8].

The most common division of osteogenesis imperfecta is the classification by Sillence from 1979; based on clinical, radiologic, and genetic criteria, he described four osteogenesis imperfecta types (OI I-IV) [10]. In 2004, Glorieux and Rauch described three further OI types (OI V-VII) that lack the COL1A1 or COL1A2 genetic mutation [11]. In 2007, Sillence’s division was expanded with another type—VIII [12]. Furthermore, the IA subtype is distinguished, which lacks changes in dentition (dentinogenesis imperfecta), and the IB subtype, in which such changes are present. OI type II is the most severe form of the disease, as fractures occur already in the uterus and in consequence, children typically die right after birth due to hypoxemia [11]. OI type III is characterized by intensified bone brittleness, slow growth, progressing scoliosis, and bone deformities, which impedes walking.

Nowadays the aim of operation procedures is reducing the number of fractures and the risk of bone deformations in the future [13,14]. Pharmacological treatment utilizes biphosphonates, the action of which consists in reducing bone resorption by inhibiting the action of osteoclasts in the mineralized bone matrix. Biphosphonate administration has been found to increase bone mineral density, reduce the number of fractures, relieve pain and musculoskeletal fatigue, and improve muscle strength and mobility, thus enhancing functioning in daily life [15,16,17]. In research conducted on a mice model with OI, the advantages of treatment with stem cells was underlined to reduce the number of fractures and increase the elasticity and neutrality of bones, and the amount of endogenous enzymes participating in ossification [15,18].

The importance of prevention programs has been underlined for the purpose of OI patient rehabilitation, as training contributes to the enhancement of physical performance and reduced subjective fatigue [19]. Prevention programs should be introduced as early as possible due to the considerable decrease of motor capabilities in people with OI III and IV after the age of 20, which derives from progressing spine deformities, more frequent use of a wheelchair, and reduced motivation for exercise [20]. Patients with OI IV experience deficit in muscle strength in the lower limbs, while the muscle strength in the upper limbs is normal [21]. In physiological terms, the primary rotation of the cervical spine is associated with lateral bend on the same side. The maximum rotation range is about 40° and is accompanied by about 28° lateral flexion [22].

During growth and development, a child’s cervical spine is subject to dynamic changes. They are associated with, among others, attaining an upright position, vertebra ossification changes in the cord tensions, and reduced ratio of the head size to the rest of the body [23]. Restriction to the active spinal mobility in children with OI can be affected by changes in the bone structure and neck pain. Thus far, few studies have been conducted that would enable the creation of standards with division by age and sex for the mobility of individual joints in children [24,25,26,27].

With age, the active cervical spinal mobility is reduced as a result of degenerative changes and reduced muscle strength [28]. The temporomandibular joints (TMJ) are the only joints in the human body that are bound in functional and anatomical terms, by the mandibular body and rami. Asymmetrical loading of these joints is the main cause of functional disorders of the masticatory apparatus [29].

The articular surface of the temporal bone is small and has a concave shape. Its shape changes with age. It is flat in children; then, it is deepened when permanent teeth appear, and then flattens again when the teeth are lost [29]. The cranium, mandible, shoulder, and cervical spine form one functional unit via articular, muscular, and fascial connections. It is very common for disorders occurring in the TMJ to lead to dysfunctions in the cervical spine and shoulder [22,30,31].

This study’s objective was to assess whether children with OI exhibit symptoms of temporomandibular disorders (TMD) and if there are differences in comparison with healthy children. The second aim was to assess whether there is relationship between the active range of motion (ROM) of the cervical spine and TMJ in comparison with healthy children. A further aim was to present the TMD prevention algorithm for children with OI. Similar studies could not be found in a literature review.

## 2. Materials and Methods

The study was carried out from November 2018 to March 2019 on the premises of the Orthopedic Clinic of the University Children’s Hospital CMUJ in Krakow and at the Department of Orthopedics and Physiotherapy in Krakow (Poland).

The study included children whose parents or legal caregivers, after reading the aim, scope, and course of the study, provided written informed consent to conduct study procedures and to process the personal data of the child in accordance with the Ordinance of the European Parliament and of the Council of EU of April 27, 2016 on the protection of individuals. The study was approved by the Ethics Committee of the Jagiellonian University Medical College (ID KBE- 1072.6120.33/2017; date of approval: 28.09.2017) and was conducted in accordance with the Declaration of Helsinki of the World Medical Association concerning ethical procedures for medical studies involving human participants. Information regarding clinical trial registration is available at www.ClinicalTrials.gov (identifier NCT03736408).

A total of 57 participants were included in the study; the study group comprised 34 children aged 3 to 18 years, 15 were female and 19 male. The mean age was 9.1 ± 3.8 years.

The basic inclusion criterion for the study was the determination of congenital OI assessed by an orthopedist with over 20 years of experience, based on clinical, radiological, and/or genetic diagnosis. The exclusion criteria included instable fractures in the cervical spine and other psychomotor disorders disabling the performance of active movements in the cervical spine, as well as age below 3 years. Children were recruited for the study during their control examination at the clinic. Table 1 presents the characteristics of the study group.

The control group included 23 healthy children, aged 3 to 15 years, of whom 8 were female and 15 male. Mean age was 9.7 years. Similar to the experimental group (one child was excluded), one child was excluded also from the study due to failure to meet the inclusion criteria. The exclusion criteria comprised skeletal system diseases and mandibular injuries requiring surgical intervention as well as age below 3 years

The research tools used in the study were a survey questionnaire containing metric data and information on the course of the disease, number and location of fractures, mean of food intake after birth, pain, and current physiotherapy. We used the Cervical Range of Motion Instrument (CROM) inclinometer, enabling a precise angle measurement of the cervical spine motion range on three planes: sagittal—flexion and extension, coronal—lateral flexion right and left, complex—rotation right and left (Figure 1). A centimeter ruler was used to assess the mandibular mobility in the range of abduction (jaw opening), protraction, and lateral mandible movements (Figure 2), as well as the Helkimo scale.

The track of the mandible and presence or absence of crackles or crepitations occurring during the performance of movement in the TMJ were also assessed according to the Diagnostic Criteria for Temporomandibular Disorders (RDC/TMD) of cracking.

The measurements were performed in a corrected seated position, without support to the rear portion of the back. Each of the participants performed the following movements: flexion and extension of the cervical spine on the sagittal plane, lateral right and left flexion on the coronal plane, and rotation right and left.

To perform the statistical analysis of the obtained data, Statistica 13.3 and Microsoft Office Excel 2007 software were used. An equal level of statistical significance was assumed for all analyses: α = 0.05.

## 3. Results

Type I OI was found in half of the study group (*n* = 17), and 35% of children had type III osteogenesis imperfecta (*n* = 12), whereas confirmed type IV OI was found only in 1% of the examined children (*n* = 4). Type II of OI was not confirmed in any of the children. Dentinogenesis imperfecta was determined in 47% of the participants. Mandible deviation to the right or left was found in 79% of children. According to their report, only 53% of the children performed regular exercises under the instruction of a physiotherapist. In the case of detecting irregularities in the TMJ, children were ordered to perform jaw-tongue coordination exercises at home.

Crackles and crepitations according to the medical history or during temporomandibular joint examination were determined in 24% of children in the study group and in 11% of children in the control group (Table 2).

During analysis, attention was paid to the flexion, extension, lateral flexions to the right and left, and right and left rotations in the cervical spine joints, as well as the scope of lateral abduction, protraction, and translation of the mandible. Distributions of the investigated parameters in both groups jointly were not compliant with a normal distribution. Mann-Whitney U nonparametric test revealed statistically significant differences between the study and control groups in the range of motion of the cervical spine in extension and rotation right and left (*p* < 0.05). Differences in the mobility in the remaining directions between the groups were not statistically significant. Characteristics of the ROM in the cervical spine in children in both groups are presented in Figure 3.

Reduced mobility of the cervical spine in all directions was found in children with OI. Increased range of motion in the temporomandibular joints relative to the individual functional norm measured for each of the participants was determined in 74% of the children in the study group and in 11% in the control group. However, in the control group, 21% experienced limitation of motion in the temporomandibular joints relative to the functional norm. Mean ranges of motion in the temporomandibular joints in both groups are presented in Table 3.

Spearman rank-order correlation showed statistically significant relationships between traits in the study group. Correlation between cervical spine extension (*p* = 0.37) and lateral flexion to the right (*p* = 0.35), and mandible abduction was found. Lateral deviation of mandible to the left correlates to a moderate degree directly proportionate with cervical spine flexion to the right (*p* = 0.36) and left (*p* = 0.35), presented in Table 4.

## 4. Discussion

The results of the present study revealed considerable differences in the mobility of the cervical spine in children with congenital osteogenesis imperfecta relative to the mean values presented in the study by Arbogast (Normal Cervical Spine Range of Motion in Children 3–12 Years Old) [25]. What is more, increased mobility of the cervical spine in children with OI type I was observed, which can be explained by the general polyarticular laxity present in the disease. On the other hand, in children with type III and IV OI, the mobility was considerably lower than in children with type I OI. A study by Engelbert showed different joint mobility for various OI types, as particularly in children with type III, the range of motion in shoulder, hip, knee, and ankle was considerably reduced, and generalized joint hypermobility was observed only in children with OI type I, which was also confirmed in the present study [3].

However, few studies to date have described normal mobility of joints in children with OI taking age into consideration. Smoląg et al. included dysfunctions of the masticatory system, yet without the assessment of the cervical spine [10]. It appears necessary to continue observations on a larger study group with OI to draw binding conclusions. Hypermobility in the temporomandibular joints was determined in the present study in 74% of the children with OI and only in 11% of children in the control group. On the other hand, in the control group, limited mobility in temporomandibular joints was more frequently observed. The data on TMD prevalence and jaw functional limitations available in the literature can serve as a source for comparison in adults patients [32] but not for children with OI.

In previous studies, few authors have focused on the problem of functional disorders of the masticatory organ in OI. In terms of genetic disorders, Młynarska-Zduniak determined that as many as 94% people with Down syndrome exhibited masticatory apparatus dysfunctions [33], but this does not refer to our study.

An interesting result of our study was the relationship observed between the extension of the cervical spine and the lateral flexion to the right, and mandibular abduction and lateral deviation of the mandible to the left and the cervical spine flexion both to the right and left in children from the study group, which may suggest compensation mechanisms in the case of lateral spine flexion to the right, and mandibular abduction and lateral displacement of the mandible to the left. No similar analysis could be found in the literature review. The results of our study suggest that the stomatognathic system in OI should be assessed in every child.

One of the strengths of our study was that we could assess children with many types of OI. To our knowledge, this study is the only study to investigate the association between the active range of motion (ROM) of the cervical spine and TMJ in OI in comparison with healthy children. A cross-sectional study by Bendixen et al. of temporomandibular disorders and dental occlusion in a population of adult OI patients showed a reduced jaw opening capacity in 8.9% of the patients with mild OI and in 25.0% of patients with moderate-severe OI [34].

Our results showed that children with congenital bone fragility had acoustic symptoms more often than did healthy children. These findings include probable key components for future attempts to develop a TMD prevention algorithm in children with OI. According to Balkefors et al. [35], painful temporomandibular disorders, physical disabilities, and pain are generally very frequent in all OI types. This is not in line with our study, as our survey questionnaire found only two cases with mild pain in TMJ in study group. Probably it is connected with generalized joint hypermobility observed only in children with OI type I, which was also confirmed in the present study [36].

Another study [37] used a different methodology with study group; however, given the adult sample and missing control group, it cannot be easily compared with our study, even if whole-spectrum OI patients were enrolled and atlantoaxial rotatory fixation was measured.

Lack of control group in previous research restricted comparisons. Our control group involved only 24 individuals acting a small sample size relative to the study group. Physiotherapy for treating TMD in OI has been understudied in the current literature, and further research is required using bigger samples to assess effective interventions for children with OI.

Efforts should be made to increase mobility in individuals with OI in terms of gait, gross motor function, daily activity, and adaptive sport. A study of functional exercise in children with OI indicated that a standardized fitness program can improve aerobic performance and muscle strength [19]. Further reports have also been published indicating the importance of promoting physical activity in children and adolescents due to the increased prevalence of obesity in patients with OI [38,39]. The results of the study by Kruger et al. [39] pointed to the mobility limitations of specific types of OI and the benefits of developing rehabilitation protocols for this population [40], which is essential to our study in terms of TMD prevention.

Studies on the topic are limited, and the theoretical foundation is poor; however, we consider this limitation as an opportunity to identify new paradigms in physiotherapy and prevention regarding TMD for patients with OI and highlight the need for further development in this area of study.

## 5. Conclusions

The present study showed that the range of mobility of the cervical spine and TMJ increased only in children with OI type I. In the remaining types (III and IV) of OI, there was limitation of motion in both the cervical spine joints and TMJ. Rehabilitation algorithms of TMD prevention in children to exclude soreness of the spine in type I OI should introduce isometric muscle strengthening of the spinal muscles, masseter muscles, and Gerry exercises for the tongue and mandible coordination. Isometric muscle strengthening of the spinal and masseter muscles, and Gerry exercises for the tongue and mandible coordination can be also used for OI types III and IV but in a lying position for the exercises of the cervical spine. Active exercises with minimal diagonal movements of the head should be performed in order to balance the work of the suboccipital muscles as well as the suprahyoid and infrahyoid muscles.

The conducted research showed that in children with OI, acoustic symptoms from the TMJ occurred more often than in healthy children; hence, the TMD prevention programs should include exercises for tongue-mandibular coordination along with stabilization exercises for the cervical spine.

## Figures and Tables

**Figure 1 ijerph-18-01076-f001:**
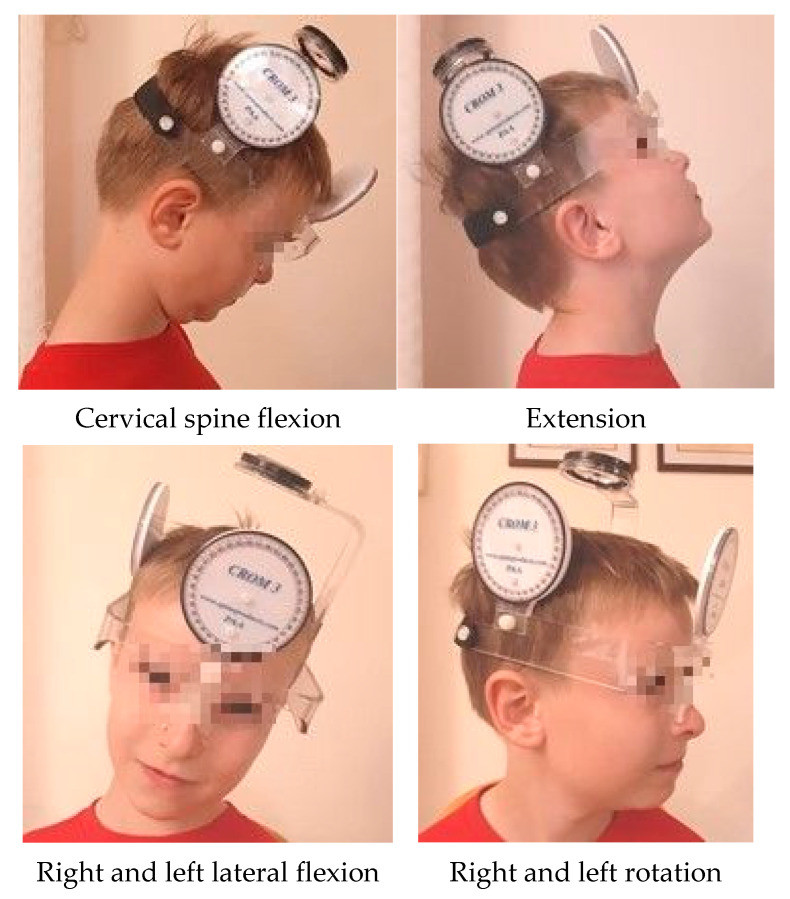
Measurement of the cervical spine motion range (Cervical Range of Motion Instrument).

**Figure 2 ijerph-18-01076-f002:**
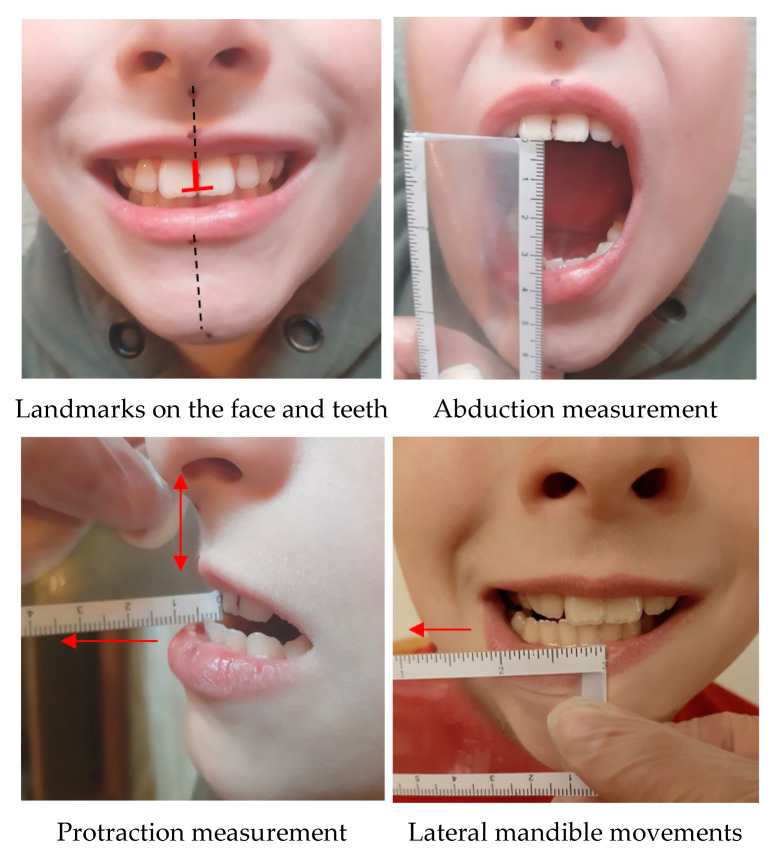
Assessment of mandibular mobility.

**Figure 3 ijerph-18-01076-f003:**
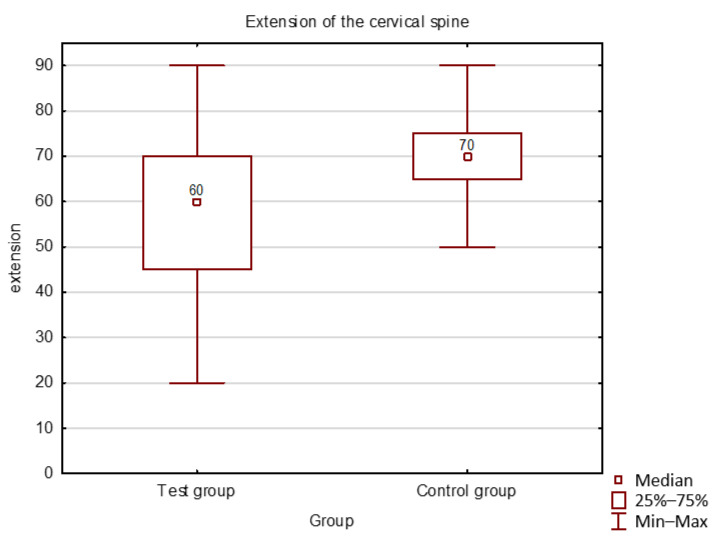
Cervical spine extension in both groups.

**Table 1 ijerph-18-01076-t001:** Characteristics of the study group in terms of mandible mobility.

Variables	Total	Female Children	Male Children
Number of participants [n]	34	15 (44%)	19 (56%)
Age [y ± SD]	9.1 ± 3.8	8.9 ± 3.7	9.3 ± 3.9
Age 3–5 years [n]	5	1 (20%)	4 (80%)
Age 6–8 years [n]	12	8 (67%)	4 (33%)
Age 10–12 years [n]	7	2 (29%)	5 (71%)
Age 13–15 years [n]	10	4 (40%)	6 (60%)
Type I OI [n]	17	6 (35%)	11 (65%)
Type III OI [n]	12	7 (53%)	5 (47%)
Type IV OI [n]	4	2 (50%)	2 (50%)
Regular physiotherapy [n]	18	9 (50%)	9 (50%)
Dentinogenesis imperfecta [n]	16	8 (50%)	8 (50%)
Deviation of the mandible to the right [n]	16	8 (50%)	8 (50%)
Deviation of the mandible to the left [n]	11	5 (45%)	6 (55%)

**Table 2 ijerph-18-01076-t002:** Incidence of crackles and crepitations in groups.

	Crackles and Crepitations	Cervical Spine Rotation to the Right
Study group	24%	62
Control group	11%	80

**Table 3 ijerph-18-01076-t003:** Mean ranges of motion in temporomandibular joints in both groups.

	Age	Jaw Opening [mm] ± SD	Functional Norm [mm]	Protrusion [mm] ± SD	Translation to the Right [mm] ± SD	Translation to the Left [mm] ± SD
Test group	3–5	37.2 ± 7	35.2	5.8 ± 2	5.9 ± 2	7.3 ± 2
	6–8	39.5 ± 4	36.0	5 ± 2	7.8 ± 2	8.3 ± 2
	9–12	36.3 ± 9	40.1	6.3 ± 3	6.3 ± 2	5.6 ± 4
	13–15	40.8 ± 7	43.3	5 ± 2	7.5 ± 2	7 ± 2
Control group	3–5	34.0 ± 2	35.0	5.3 ± 1	7.7 ± 2	7.0 ± 0
	6–8	39.3 ± 9	39.0	6.0 ± 5	7.3 ± 3	7.3 ± 4
	9–12	41.8 ± 4	44.1	5.4 ± 2	8.3 ± 1	8.8 ± 3
	13–15	44.0 ± 5	46.0	4.2 ± 3	8.4 ± 4	9.6 ± 5

**Table 4 ijerph-18-01076-t004:** Correlations between studied parameters.

Variables	Jaw Opening	Protrusion	Translation to the Right	Translation to the Left
flexion	0.13	−0.01	0.12	0.25
extension	0.37	0.13	0.13	0.10
flexion to the right	0.35	0.11	−0.05	0.36
flexion to the left	0.25	0.04	0.11	0.35
rotation to the right	0.33	−0.04	0.08	0.03
rotation to the left	0.31	−0.09	−0.02	0.14

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
