# Peer review of "Temporomandibular Joint and Cervical Spine Mobility Assessment in the Prevention of Temporomandibular Disorders in Children with Osteogenesis Imperfecta: A Pilot Study"

_ijerph, 2021, doi:10.3390/ijerph18031076_

Round 1
Reviewer 1 Report
This study is poorly written and is absolutely not meet the standard for publication at the present format.
Title - Title is not clear, and it has to be regarded as a pilot study due to a convenient small sample size. The aim of this observational study was not clear, they should check their aim to see if they want to examine the relationship between the active range of motion of the cervical spine, the mobility of temporomandibular joints and presentation of a prevention algorithm from temporomandibular disorders in children with Osteogenesis Imperfecta.
Aim - Aim of the It is not clearly state which are the dependent and independent variables they wanted to examine. Or the authors want to compare the independent variables such as active range of motion of the cervical spine of the children with and without Osteogenesis Imperfecta.
Method - The method should be rewritten according to the aim
Results - Table 1 should be revised to properly summarise the data
Figure 1 to 3 are unnecessary and can be resent in a simple table.
Discussion - The discussion is empty and short, and does not show the authors understand the tpic of their research.
Conclusion - Conclusion should address the aim of the study.
Reviewer 2 Report
Osteogenesis imperfecta (OI) is a monogenic bone fragility disorder that usually is caused by mutations in one of the two genes coding for collagen type I alpha chains, COL1A1 or COL1A2. (OI) is characterized by a number of deviations in the orofacial region.
The present study showed that the range of mobility of the cervical spine and
temporomandibular joints is increased only in children with I type of osteogenesis imperfecta
The guide should have been followed STROBE Guidelines
You must add the limitations
point out strengths
The figures 1 and 3 should be eliminated
Because the occlusion was not registered
Reviewer 3 Report
Dear Authors
I should like to thank you for give me an opportunity to consider this work for publication. You well done. The study is well conducted and had a solid methodological basis. Due to the rarity of OI, this study will aid research in this area. However, I am very sorry not to be able to see the image of functional assessment. The paper would has gained much more prestige.
Best regards
Reviewer 4 Report
We read with great interest the manuscript entitled “Temporomandibular joint and cervical spine mobility assessment in the prevention of Temporomandibular disorders in children with osteogenesis imperfecta” aiming to assess whether there is a relationship between the active range of motion of the cervical spine, the mobility of temporomandibular joints due to differential diagnosis in children with Osteogenesis Imperfecta. The results showed that in children with congenital bone fragility more often than in healthy children, there were acoustic symptoms from the TMJ and a correlation was found between the mobility of the cervical spine and temporomandibular joints in the study group.
Please find hereafter the corrections needed prior to manuscript resubmission. Some required revisions are absolutely mandatory before resubmission, and are related to the methodology of the presented work.
- Title: please decide if “Temporomandibular disorders” and “osteogenesis imperfecta” should be written with capital letters. If you decide to keep them in capital letters please correct this throughout the manuscript text. For example, in the abstract “osteogenesis imperfecta” is written with capital letter and in the title no.
- Abstract: “Osteogenesis Imperfecta, is a heterogeneous group of hereditary disorder …..“, Please correct disorder with “disorders”
- Introduction: “This mutation can be inherited from one of the parents or it may appear in a child de novo “Please correct the style of “de novo”, it should be in italics.
- Introduction: “The most common division of osteogenesis imperfecta is the classification of Sillence from 1979. Based on clinical, radiologic and genetic criteria, he described four osteogenesis imperfecta types (OI I-IV) [10]. In 2004, Glorieux and Rauch described three further OI types (OI V-VII), which lack the COL1A1 or COL1A2 genetic mutation. In 2007, the Sillence’s division was expanded with another type - VIII [11]. Furthermore, the IA subtype is distinguished, which lacks changes in dentition (dentinogenesis imperfecta) and IB subtype, in which such changes are present. Type II is the most severe form of the disease, believed to be lethal until recently. Fractures occur already in the uterus, with severe deformities of long bones and ribs present. Cranial bones are not appropriately mineralized. Children typically die right after the birth due to hypoxemia [12]. OI type III is characterized by intensified bone brittleness, slow growth, progressing scoliosis and bone deformities, which impedes walking. Nowadays, small children in which bone deformities have not yet occurred are subject to operations. The procedures aim at reducing the number of fractures and the risk of bone deformations in the future [13,14]. Pharmacological treatment utilizes biphosphonates, the action of which consists in reducing bone resorption by inhibiting the action of osteoclasts in the mineralized bone matrix. As a result of biphosphonate administration, increased bone mineral density and vertebra height, reduced number of fractures, relief of pain and musculoskeletal fatigue, improved muscle strength and mobility are observed, and thus enhanced functioning in daily life [15, 16, 17]. Nowadays, the advantages of treatment with stem cells is underlined. In the research conducted on a mice model with OI, injection of human chorionic stem cells reduced the number of fractures, increased elasticity and neutrality of bones, as well as the amount of endogenous enzymes participating in ossification, suggesting that the placenta can be a practical source of stem cells in OI treatment [15, 18]. “ This section is very interesting but I think it isquite too long, so please try to summarize this part of the text.
- Introduction:”Importance of prevention programs has been underlined for the purpose of OI patient rehabilitation, because training contributes to their enhancement of physical performance (determined with VO2) “ Please explain VO2.
- Introduction: “The study objective was to assess whether children (aged 3 to 15 y.o.) with osteogenesis imperfecta (OI) exhibit symptoms of Temporomandibular disorders (TMD) and whether there are differences in the mobility of the cervical spine and temporomandibular joints in comparison with healthy children, as well as if a relationship exists between active range of mobility (ROM) of cervical spine and the mobility of temporomandibular joints in children with osteogenesis imperfecta and in healthy children. A further aim was to present the TMD prevention algorithm in children with OI. Similar studies could not be found in the literature review. “ Please move the age range of the enrolled patients to results section, it shouldn’t be stated here. Moreover, once the authors defined the acronyms used in the text, it shouldn’t be repeated.
- Material and Methods: “The study was carried out from November 2018 through March 2019 on the premises of the Orthopedic Clinic of the University Children’s Hospital CMUJ in Krakow and at the Department of Orthopedics and Physiotherapy in Krakow after receiving written approval of the Hospital Management. “ Please add the Country.
- Material and Methods: Mandatory is the number of Local Ethical Committee approval. Please add it to the text and please add the date of the approval.
- Material and Methods: there is a big criticism related to the sample size. A) no sample size calculation is provided ; B) control group should be equal or twice as large as the study group. I honestly think this is a big limitation to this study.
- Materials and Methods: “The research tools used during the study were….”Please provide adetailed description of all the used tools.
- Materials and Methods: “The track of the mandible and presence or absence of crackles or crepitations occurring during the performance of movement in temporomandibular joints was also assessed. “ Please correct “was “ with “were”.
- Discussion: Please add to the text one section entitled “Limitations” and state precisely all the limitations to this study.
Round 2
Reviewer 1 Report
The authors revised the manuscript but the discussion is still short and empty.
Reviewer 4 Report
Thank you for correctly implementing the needed changes to the text.
No further changes are required.